# Obesity-Altered Adipose Stem Cells Promote ER^+^ Breast Cancer Metastasis through Estrogen Independent Pathways

**DOI:** 10.3390/ijms20061419

**Published:** 2019-03-20

**Authors:** Rachel A. Sabol, Adam Beighley, Paulina Giacomelli, Rachel M. Wise, Mark A. A. Harrison, Ben A. O’Donnnell, Brianne N. Sullivan, Jacob D. Lampenfeld, Margarite D. Matossian, Melyssa R. Bratton, Guangdi Wang, Bridgette M. Collins-Burow, Matthew E. Burow, Bruce A. Bunnell

**Affiliations:** 1Center for Stem Cell Research, Tulane University School of Medicine, New Orleans, LA 70112, USA; rsabol@tulane.edu (R.A.S.); abeighle@tulane.edu (A.B.); pgiacomelli@tulane.edu (P.G.); rwise@tulane.edu (R.M.W.); mharri26@tulane.edu (M.A.A.H.); bodonne1@tulane.edu (B.A.O.); bsulliv7@tulane.edu (B.N.S.); jlampenfeld@tulane.edu (J.D.L.); 2Tulane Brain Institute, Tulane University, New Orleans, LA 70118, USA; 3Department of Hematology and Oncology, Tulane University School of Medicine, New Orleans, LA 70112, USA; mmatossi@tulane.edu (M.D.M.); bcollin1@tulane.edu (B.M.C.-B.); mburow@tulane.edu (M.E.B.); 4College of Pharmacy, Xavier University, New Orleans, LA 70125, USA; mbratton@xula.edu (M.R.B.); gwang@xula.edu (G.W.); 5Tulane Cancer Center, Tulane University, New Orleans, LA 70112, USA; 6Department of Pharmacology, Tulane University, New Orleans, LA 70112, USA; 7Division of Regenerative Medicine, Tulane National Primate Research Center, Covington, LA 70433, USA

**Keywords:** adipose stem cells, breast cancer, obesity, metastasis, estrogen receptor

## Abstract

Adipose stem cells (ASCs) play an essential role in tumor microenvironments. These cells are altered by obesity (obASCs) and previous studies have shown that obASCs secrete higher levels of leptin. Increased leptin, which upregulates estrogen receptor alpha (ERα) and aromatase, enhances estrogen bioavailability and signaling in estrogen receptor positive (ER^+^) breast cancer (BC) tumor growth and metastasis. In this study, we evaluate the effect of obASCs on ER^+^BC outside of the ERα signaling axis using breast cancer models with constitutively active ERα resulting from clinically relevant mutations (Y537S and D538G). We found that while obASCs promote tumor growth and proliferation, it occurs mostly through abrogated estrogen signaling when BC has constitutive ER activity. However, obASCs have a similar promotion of metastasis irrespective of ER status, demonstrating that obASC promotion of metastasis may not be completely estrogen dependent. We found that obASCs upregulate two genes in both ER wild type (WT) and ER mutant (MUT) BC: *SERPINE1* and *ABCB1*. This study demonstrates that obASCs promote metastasis in ER WT and MUT xenografts and an ER MUT patient derived xenograft (PDX) model. However, obASCs promote tumor growth only in ER WT xenografts.

## 1. Introduction

Breast cancer (BC) is the most common cancer and the second leading cause of cancer death in women [1]. An estimated 246,000 patients were diagnosed with new cases of BC in 2016 [1]. Among the many factors that influence BC, obesity has been shown to increase the rates of many types of cancer [2]. Studies have shown a positive correlation between adult body mass index (BMI) and the incidence of postmenopausal breast cancer, specifically ER^+^ breast cancer [2,3]. Obesity has also been found to increase both breast cancer recurrence and mortality [4]. Obesity is defined as having a BMI of ≥ 30 kg/m^2^ [5]. Rates of obesity have been increasing in the United States since the 1970s [6]. It is estimated that 21% of the world’s female population will be obese as of 2025, illustrating a critical need to interrogate the relationship between obesity and BC [7]. Obesity can promote breast cancer through many mechanisms including alterations to tumor biology to promote a more aggressive cancer phenotype or metabolic reprogramming. Many metabolic intermediates such as pyruvate kinase M2, sterol regulatory element-binding protein 1c, and peroxisome proliferator-activated receptors γ (PPARγ) are differentially regulated in obesity [8,9,10]. These factors have been implicated in reprogramming in the tumor microenvironment (TME) and have been shown to promote breast cancer proliferation and migration and even alter epigenetics leading to increased cancer incidence [11,12,13]. Data demonstrates targeting metabolic reprogramming with agents, such as statins or thiazolidinediones, can be as chemoprevention in rats [14,15]. It is clear that obesity alterations in the TME can alter cancer biology. 

Adipose stem cells (ASCs) play a central role in the TME. They have been shown to support angiogenesis via recruitment of blood resident endothelial progenitors and promote inflammation [16]. ASCs effectively increase tumor growth, as well as the motility and invasive capacity of cancer cells via the stromal-derived factor 1/chemokine receptor type 4 (SDF-1/CXCR4) axis [17,18]. ASCs also induce an epithelial-to-mesenchymal (EMT) transition in cancer cells through platelet derived growth factor (PDGF) signaling [19], and a cancer stem-like phenotype in breast cancer through adipsin [20]. Additionally, cancer cells can induce a change in ASCs to cancer-associated fibroblasts (CAFs), which, in turn, increase secretion of factors that further enhance tumor proliferation, invasion, and metastasis [18,21]. Obesity increases the rate of CAF conversion, leading to enhanced proliferation and increased invasive capability of cancer cells [22]. 

It has previously been shown that ASCs from obese donors promote BC tumor growth and metastasis when compared to those cultured alone or with lean donor ASCs (BMI < 25) (lnASCs) [23]. Studies have demonstrated that obesity-altered ASCs (BMI > 30) (obASCs) increase the proliferation and tumor size of estrogen receptor positive (ER^+^) BC and increase lung and liver metastasis. Leptin, an adipokine abundantly secreted by obASCs relative to lnASCs, promotes ER^+^BC growth and metastasis by increasing expression of ERα and aromatase [24]. Previous reports show that knocking down leptin in obASCs reduces but does not entirely ameliorate the effect obASCs have on each of these processes. This suggests that obASCs may also act through non-estrogen pathways to promote ER^+^BC tumorigenesis and metastasis [23]. In this study, clustered regularly interspaced short palindromic repeats-CRISPR-associated protein 9 (CRISPR-Cas9) generation of ER^+^BC cell line MCF7 with a gnomically encoded Y537S mutation in the estrogen receptor alpha (*ESR1*) gene was used [25], and patient-derived xenograft models in which the patient developed the Y537S or D538G mutations were used. D538G and Y537S mutations in ERα result in constitutive ERα activity [26]. These are clinically relevant mutations that develop *de novo* in BC and result in BC that is resistant to endocrine therapies that target ERα [27]. By employing these genetically modified breast cancer cells (BCCs), the effect(s) obASCs have on ER^+^BC outside of the ERα-leptin signaling axis can be investigated. 

## 2. Results

### 2.1. Obesity-Altered Adipose Stem Cells Promote Metastasis but Not Tumor Growth of Breast Cancer with Mutant ERα

Previous studies have reported that obASCs promote growth and metastasis of MCF7 xenografts [24]. We demonstrate that obASCs enhance the growth of MCF7+estrogen pellet xenografts and have increased metastatic index (Appendix A); however, MCF7-Y537S xenografts and WHIM20 patient-derived xenografts that have the Y537S mutation do not have enhanced growth in the presence of obASCs compared to lnASCs or control xenografts with no stem cells (Figure 1A). obASCs increase the metastatic index significantly in MCF7-Y537S xenografts with 3.30% ± 0.76 (Mean ± standard error of the mean (SEM)) area occupied by metastases in the obASC group compared to 0.69% ± 0.16 with lnASCs and 1.04% ± 0.21 with control (Figure 1B). Similarly, WHIM20 PDX demonstrated 2.36% ± 0.09 area occupied by metastases in the obASC group compared to 1.70% ± 0.08 with lnASCs and 0.95% ± 0.27 with control (Figure 1B). Flow cytometry was used to evaluate the presence of circulating tumor cells in the WHIM20 PDX model and found that mice with PDX tumors grown with obASCs had no significant difference in human (HLA1^+^) circulating tumor cells (CTCs); however CTCs from tumors grown with obASCs demonstrated a trend of enrichment for the breast cancer stem cell markers CD44^+^CD24^−^ (Figure 1C). MCF7-Y537S and WHIM20 xenografts+estrogen pellets show a similar trend to xenografts without estrogen pellets where there is no effect of obASCs on tumor growth, but obASCs promote tumor metastasis and circulating tumor cells (CTCs) (Appendix A).

### 2.2. In Vitro obASCs Promote Proliferation and Migration of ER WT and ER MUT Cells

To evaluate in vitro the effects observed in vivo, conditioned media (CM) from ASCs was used and measured BCC proliferation over time. Secreted factors from obASCs promote proliferation of ER^+^BCCs in vitro. While obASC conditioned media (CM) had a greater effect on the proliferation of MCF7 than MCF7-Y537S, a trending increase in proliferation was observed when estrogen is constitutively active (Figure 2A). This demonstrates that obASCs exert some effect through the estrogen receptor on ER-dependent breast cancer cells. CM from obASCs did significantly increase the proliferation of PDX-derived cells with constitutive ER activity (Figure 2A). To study the metastatic phenotype in vitro, a migration assay was used and indicated that obASCs significantly promoted migration of ER^+^BCCs and a PDX-derived cell line irrespective of ER WT or MUT status (Figure 2B). Specifically, in ER WT BC (MCF7) an average of 60.7 cells ± 5.0 (Mean ± SEM) migrated to obASCs compared to 4.3 ± 3.8 cells to lnASCs, and 10.0 ± 7.2 cells migrated to CCM. ER MUT BC cell line (MCF7-Y537S) showed a similar trend with an average of 32.3 cells ± 5.8 (Mean ± SEM) migrated to obASCs compared to 11.0 ± 2.5 cells to lnASCs, and 9.7 ± 1.5 cells migrated to CCM, and ER MUT PDX derived cells (33.3 ± 3.8 to obASCs, 5.7 ± 0.9 to lnASCs, and 6.7 ± 2.4 to CCM) (Figure 2B). This further supports the hypothesis that obASCs are promoting metastasis in the TME outside of the estrogen-signaling axis. 

### 2.3. Regulation of Breast Cancer Related Genes in ER WT and ER MUT Cells by obASCs

The Qiagen RT^2^ Profiler™ PCR Array Human Breast Cancer was used to determine genes and pathways altered in ER WT and MUT cells after 96-h Transwell co-culture with obASCs. A cutoff of 2-fold expression change was established, which resulted in the identification of 22 genes upregulated by obASCs in ER WT cells (MCF7), and 9 genes were downregulated by obASCs. In comparison, ER MUT (PDX-derived WHIM43 cells) had 5 genes upregulated by obASCs and 4 genes downregulated after co-culture with obASCs (Figure 3, Table 1). Of the 22 upregulated genes, obASCs upregulated estrogen-related genes that have been shown to promote tumor growth, such as *ESR2*; as well as cell-cycle-related genes associated with proliferation, such as *CDKN1C* and CDKN2A. ER WT and MUT cells upregulated 2 genes: *SERPINE1* (2.51 fold in ER WT and 4.47 fold in ER MUT) and *ABCB1* (6.48 fold in ER WT and 4.99 fold in ER MUT) (Figure 3, Table 1). In light of our results showing that obASCs promote metastasis but not tumor growth in ER MUT BC, it is likely that these overlapping gene changes are associated with the increased metastasis observed in ER WT and ER MUT BC when exposed to obASCs. These two genes could be a common mechanism outside of the estrogen axis through which obASCs promote metastasis of ER WT and ER MUT tumors.

## 3. Discussion

Obesity is recognized as a leading preventable cause of cancer [28]. Obesity has also been shown to alter the way ASCs interact with cancer cells within the TME [29]. However, there is little research on obesity-specific cancer therapies that target the TME, despite the crucial role obesity-associated cytokines play in cancer development and progression. The current study aimed to reveal cross-talk between obASCs and ER^+^BCCs outside of estrogen-dependent signaling and to identify novel pathways through which obesity promotes BC metastasis.

Studies have demonstrated that ASCs promote breast cancer cell migration, stemness and promote angiogenesis [30,31,32]. ObASCs have been shown to promote tumor growth and metastasis of MCF7 through increased secretion of leptin, which upregulates aromatase and ERα [24]. In this study, we show that obASCs exert effects on BC through non-estrogen signaling pathways. We found that obASCs promote metastasis, but not tumor growth of ER^+^BC with mutations in ERα that result in constitutive ERα activity. We found that obASCs promote a metastatic phenotype *in vitro*, which correlates with increased circulating tumor cells and circulating cancer stem-like cells in PDX models and increased lung metastases in cell line and PDX models. To evaluate genes and pathways commonly activated in ER WT and ER MUT cells we used an array of 84 breast cancer- related genes. Some genes were differentially regulated in ER WT and ER MUT cells; however, only two genes were upregulated by obASCs in bot ER WT and ER MUT: *ABCB1* and *SERPINE1*. 

*ABCB1*, also known as multidrug resistance 1 or P-glycoprotein, is an efflux transporter with an ATP binding cassette [33,34]. Overexpression of *ABCB1* in breast cancer is associated with poor response to first-line chemotherapies because *ABCB1* can efflux many drugs used in the treatment of breast cancer such as taxanes, anthracyclines, and vinca alkaloids [35]. There are many different signal transduction pathways and transcription factors that can lead to ABCB1 transcription and ultimately chemoresistant tumors including: *Ras* [36,37,38], cyclic adenosine monophosphate (cAMP)/ Protein kinase A (PKA) pathway [39,40], protein kinase C [41,42], phosphatase and tensin homologue (PTEN) [43,44], phosphatidylinositol 3-kinase (PI3K)/protein kinase B (AKT) pathway [45,46], and p53 [47,48]. These signaling pathways could be activated by any number of known secreted growth factors and cytokines from obASCs to upregulate *ABCB1*. *ABCB1* is not known to play a role in the metastatic phenotype of breast cancer but is associated with a more aggressive cancer that is drug resistant leading to worse outcomes. 

*SERPINE1*, also known as plasminogen activator inhibitor-1, which was also upregulated in both ER WT and ER MUT BC by obASCs is associated with tumor progression and invasion [49]. *SERPINE1* is an inhibitor of urokinase plasminogen activator (*uPA*), which is itself an extracellular matrix-degrading protease associated with cancer invasion [50,51]. Based on its ability to suppress *uPA*, it was previously hypothesized that *SERPINE1* would be tumor inhibitory; however, studies have now demonstrated that *SERPINE1* plays a role in neoangiogenesis in the tumor microenvironment and thereby plays a role in tumor progression, invasion, and metastasis [52,53]. *SERPINE1* in keratinocytes in a wound healing environment have been deemed the “molecular switch” from proliferation to migration by Simone et al. [54]. Because obASCs upregulated *SERPINE1* in ER WT and MUT breast cancer and promoted metastasis, but not tumor growth in both xenograft models and a PDX model, we hypothesize that *SERPINE1* could be a key mediator in obesity-altered ASCs promotion of metastasis. Future studies are needed to thoroughly investigate this hypothesis as well as evaluate the factors secreted by obASCs that upregulate *SERPINE1* to develop therapeutic strategies to block this obesity-mediated promotion of metastatic disease.

## 4. Materials and Methods 

### 4.1. Human Subjects

All protocols were reviewed and approved by the Pennington Biomedical Research Center Institutional Review Board and all human participants provided written informed consent (PBRC #23040 approved in December, 2011) (LaCell, New Orleans, LA, USA). Human ASCs were isolated from 12 Caucasian females (2 groups, 6 donors per group) undergoing elective liposuction procedures, as previously described [24]. The mean BMI for each of the two donor groups was as follows: Obese (32.7 ± 3.7) and Lean (22.7 ± 1.9). The mean age of the subjects for each group of donors was as follows: Obese (42.5 ± 8.9) and Lean (38.8 ± 7.0). No statistical significance in age was observed between the donor groups.

### 4.2. Cell Culture

ASCs were isolated, cultured, and characterized as previously described [24]. MCF7 cells were purchased from American Type Culture Collection (ATCC; Manassas, VA, USA). MCF7-Y537S cells were obtained from the Department of Surgery and Cancer at Imperial College London (London, England) and cultured as previously described [25]. Cells were cultured in complete culture media (CCM), which consisted of α-minimal essential media (αMEM; Gibco; Grand Island, NY, USA), 10% fetal bovine serum (Atlanta Biologicals, Lawrenceville, NJ, USA) 100 units per mL penicillin/100 ug/mL streptomycin (P/S; Gibco), and 2 mM L-Glutamine (Gibco). Cells were grown at 37 °C with 5% humidified CO^2^, CCM was changed every three to four days, and split 1:4 to 1:6 when they reached 90% confluency as previously described [24].

### 4.3. RT-qPCR

BCCs (5 × 10^4^) were plated in the bottom of a 6-well plate (Nunc, ThermoFisher, Waltham, MA, USA) and six pooled donors of obASCs (BMI > 30) or lnASCs (BMI < 25) were seeded at a density of 5 × 10^4^ cells in a 0.4 µm pore Transwell (Corning Inc., Corning, NY, USA). Cells were allowed to attach overnight. After 24 h, Transwell inserts containing ASCs were transferred to wells harboring BCCs for 96 h. RNA was isolated using Qiazol (Qiagen, Valencia, CA, USA) and an RNeasy Mini Kit (Qiagen). RNA was converted to cDNA using RT^2^ First Strand Kit (Qiagen). RT^2^ Profiler™ PCR Array Human Breast Cancer with RT^2^ SYBR Green qPCR Mastermix (Qiagen) was used to identify breast cancer genes/pathways upregulated by obASCs. 

### 4.4. Conditioned Media Proliferation Assay 

Lean and obese ASCs plated on a 150 mm^2^ dish were allowed to reach 70% confluence. Plates were washed with sterile phosphate buffered saline (PBS) and the medium was replaced with serum-free αMEM for 24 h. Media was collected and filtered through a cell strainer (0.2 μm nylon mesh; Fisher Scientific, Hampton, NH, USA) to remove cellular debris. BCCs were plated at 200 cells per well of a 96-well plate in triplicate in CCM and allowed to adhere overnight. Cells were then washed with PBS and 200 μL of lean or obese ASC CM, or serum-free αMEM was added. Proliferation assay was conducted with 10% Alamar blue reagent (Invitrogen, Carlsbad, CA, USA) per manufacturer’s instructions. Proliferation quantification was done by measuring relative fluorescent units (RFU) (excitation 530–560 nm; emission 590 nm). 

### 4.5. Migration Assay

CCM or 0.5 × 10^6^ ASCs in CCM were plated in the bottom of a 6 well plate and allowed to adhere overnight. 0.5 × 10^6^ breast cancer cells were seeded in Transwells (0.4 μm pore; Corning Inc.) and allowed to adhere overnight. After 24 h Transwells were transferred to wells with CCM or ASCs in CCM and cultured for three days. Transwells were then fixed and stained with Crystal Violet (3% in methanol) for 30 min, washed with deionized water, and migrated cells were counted manually. 

### 4.6. Orthotopic Xenografts 

Four to six-week-old severe combined immunodeficiency (SCID)/beige (CB17.Cg-PrkdcscidLystbg-J/Crl) ovariectomized female mice were obtained from Charles River Laboratories (Wilmington, MA, USA). Mice were divided into six groups of five animals: MCF7-Y357S only, MCF7-Y357s plus obASCs (*n* = 6 donors), MCF7-Y357S plus lnASCs (*n* = 6 donors), MCF7-Y357S plus estradiol, MCF7-Y357S plus obASCs and estradiol (*n* = 6 donors), and MCF7-Y357S plus lnASCs and estradiol (*n* = 6 donors). Where indicated, estradiol pellets were implanted subcutaneously in the lateral area of the neck (0.72 mg, 60-day release; Innovative Research of America, Sarasota, FL, USA) as previously described [24]. 

MCF7-Y357S cells (10^6^) alone or MCF7-Y357S cells (10^6^) in combination with ASCs (10^6^) suspended in a total volume of 50 μL of sterile PBS were mixed with 100 μL of reduced growth factor Matrigel (BD Biosciences, Bedford, MA, USA). Cells were injected subcutaneously into the fifth mammary fat pad on both sides as previously described [24]. All procedures in animals were performed under anesthesia using a mixture of isoflurane and oxygen delivered by a nose cone. Tumor size was measured every three days using digital calipers and calculated as previously described [24]. At necropsy, animals were euthanized by cervical dislocation after exposure to CO_2_. Lungs were removed and fixed in 10% neutral buffered formalin and paraffin embedded for metastatic analysis. All procedures involving animals were conducted in compliance with State and Federal law, standards of the United States Department of Health and Human Services, and guidelines established by Tulane University Institutional Animal Care and Use Committee (IACUC). All protocols were approved by the Tulane IACUC (Protocol 4299R. Approved in December, 2013). 

### 4.7. Patient-Derived Xenografts 

The PDX models used in this study WHIM20 (isolated from a patient who developed the Y537S mutation) and WHIM43 (D538G mutation) were obtained from Washington University in St. Louis (Horizons Discovery Group, Waterbeach UK). All animal procedures were reviewed and approved by Tulane University IACUC. SCID/beige (CB17.Cg-Prkdc^scid^Lyst^bg-1^/Crl) 4–6-week-old female mice were obtained from Charles River Laboratory. Intact tumor pieces were removed and sliced with a scalpel to 3mm x 3mm and coated with 100 uL phenol-free growth factor reduced Matrigel (BD Biosciences). Control groups had PDX tumor coated in Matrigel and implanted bilaterally in the mammary fat pads under isoflurane and oxygen. In ASC groups, 10^6^ pooled donors (*n* = 6) of lnASCs or obASCs were resuspended in Matrigel and coated the tumor. Where indicated, estradiol pellets were implanted subcutaneously in the lateral area of the neck (0.72 mg, 60-day release; Innovative Research of America). Tumors were implanted into the fifth mammary fat pad bilaterally under isoflurane and oxygen anesthesia delivered by nose cone and animals were given 5mg/kg/day meloxicam for three days post-surgery. Tumors were measured by digital caliper every three to four days. At endpoint (tumors reach 750–1000 mm^3^) blood and lungs were collected for analysis. PDX derived cells were cloned out from PDX tumors. 

### 4.8. Flow Cytometry

To identify circulating tumor cells whole blood was collected with 0.5 M EDTA (Gibco). Samples were incubated with 0.008% NH_4_CL (ThermoFisher) for red blood cell lysis and washed with PBS. Cells were then blocked with 1% bovine serum albumin (BSA) and 1% anti-CD16/anti-CD32 (EBioscience, ThermoFisher) in PBS and stained with antibodies against HLA1 (Invitrogen), CD24 (EBioscience), CD44 (EBioscience). Samples were analyzed with a Gallios Flow Cytometer (Beckman Coulter, Brea, CA, USA) with Kaluza software (Beckman Coulter). A minimum of 10,000 events were captured and analyzed. 

### 4.9. Statistical Analysis 

The analysis was performed using Prism (Graphpad Software, San Diego, CA, USA). All values are presented as means ± standard error. Statistical differences among two or more groups were determined by ANOVA, followed by post-hoc Tukey tests versus the respective control group. Statistical differences between two groups were performed by Student’s t-test. Statistical significance was set at *p* < 0.05. 

## 5. Conclusions

ObASCs promote tumor growth and metastasis of ER WT BC, and promote metastasis of ER MUT BC. These data demonstrate that there are independent pathways promoted by obASCs that affect tumor growth and metastasis. The pathways that promote tumorigenesis are ER-dependent; however, in BC where these pathways are constitutively activated obASCs promote metastasis but have no effect on tumor growth. When 84 key breast cancer-related genes were evaluated in BCCs after Transwell co-culture with obASCs we found that obASCs upregulate more genes in ER WT than ER MUT cells. Interestingly, there were two genes upregulated in both cell types: ABCB1, a gene associated with multidrug resistance, and SERPINE1, a gene associated with an invasive metastatic phenotype. Future studies should aim to investigate the dependency of obASC promotion of metastasis on SERPINE1 expression as well as investigate the expression of SERPINE1 in human tumors from lean and obese women to see if these findings are translational. 

## Figures and Tables

**Figure 1 ijms-20-01419-f001:**
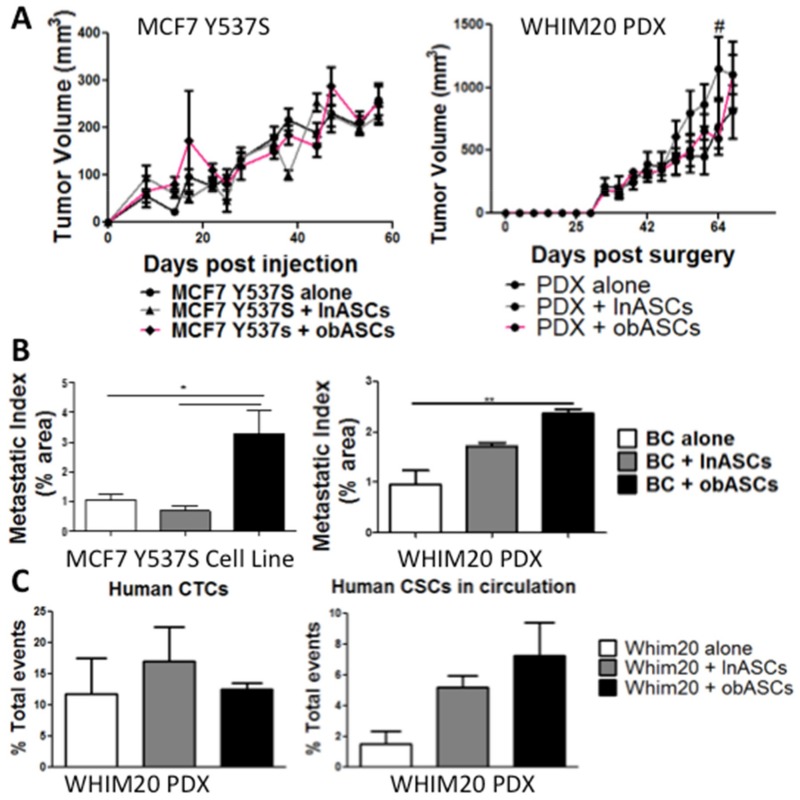
obASCs promote metastasis but not tumor growth of constitutively active ERα xenograft models—MCF7-Y537S and WHIM20 PDX (**A**) Tumor volume was tracked over time. (Day of injection = Day 0). There is no change in tumor volume when BC was implanted in the presence of lnASCs or obASCs compared to BC alone except lnASCs compared to control WHIM20 tumor volume at day 60 (# −ln *vs*. ctrl *p* < 0.05). Caliper measurements were taken every three to four days until the tumor volume reached 750–1000 mm^3^. Values reported are the mean (*n* = 5 mice/group). Data were analyzed using two-way analysis of variance (ANOVA) and a Bonferroni post-test. (**B**) Area of the lung occupied by metastasis (metastatic index) was evaluated at the endpoint. Groups, where BC was implanted with obASCs, had higher levels of metastasis compared to BC alone or grown with lnASCs. Data were analyzed using one-way ANOVA and Tukey post-test. Bars, ± SEM. * *p* < 0.05, ** *p* < 0.01. (**C**) Circulating tumor cells were analyzed in animals harboring patient-derived xenograft (WHIM20) at endpoint using flow cytometry. There was no change in human (HLA1^+^) cells across groups; however, analysis of circulating tumor cells enriched for the cancer stem cell marker CD44^+^CD24^−^ was increased in PDX+obASCs compared to PDX alone. Data were analyzed using one-way ANOVA and Tukey post-test and no significant difference was found. Bars, ± SEM.

**Figure 2 ijms-20-01419-f002:**
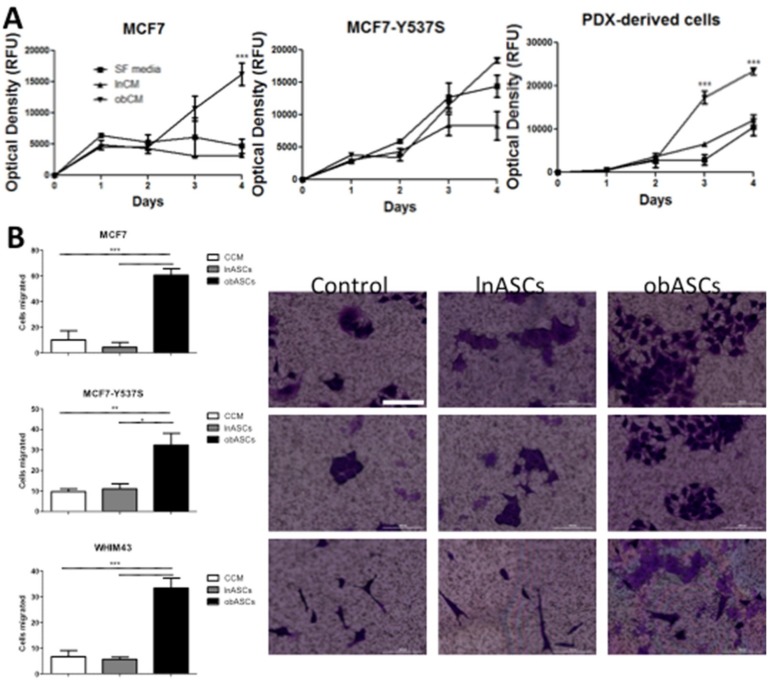
In an estrogen-depleted environment, obASCs promote proliferation and migration of ER WT and ER MUT BCCs in vitro. (**A**) Conditioned media collected from ASCs after 24 h promotes proliferation of ER WT (MCF7) and ER MUT with constitutively active ERα (MCF7-Y537S and PDX-derived) cells. Data were analyzed using two-way ANOVA and Bonferroni post-test. *** *p* < 0.001 (**B**) obASCs promotes increased migration of MCF7, MCF7-Y537S and PDX derived cells (WHIM43) through a 0.4 um membrane. Scale bar represents 100 μm. Data were analyzed using one-way ANOVA and Tukey post-test. Bars, ± SEM. * *p* < 0.05, ** *p* < 0.01, *** *p* < 0.001. Values reported are the mean of three independent experiments each performed in triplicate.

**Figure 3 ijms-20-01419-f003:**
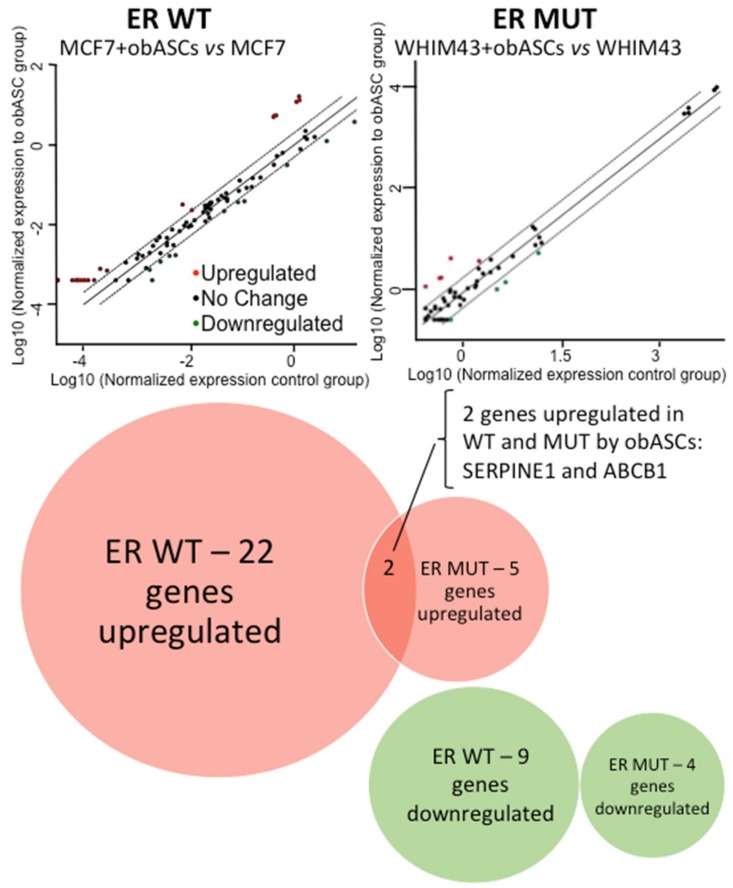
PCR array demonstrates that obASCs upregulate two common genes in ER WT and ER MUT cells. PCR array demonstrates that obASCs upregulate above a cutoff of 2x: 22 BC related genes in ER WT (MCF7) cells compared to 5 genes ER MUT (WHIM43) and downregulated 9 BC related genes in ER WT compared to 4 in ER MUT. Two genes were upregulated in both ER WT and ER MUT: SERPINE1 and ABCB1. PCR arrays were conducted (*n* = 1) for each condition.

**Table 1 ijms-20-01419-t001:** PCR array fold changes of genes upregulated (red arrow) and downregulated (green arrow) by obASCs in ER WT (MCF7) and ER MUT (PDX-derived WHIM43) cells. This table demonstrates the specific genes and fold changes in gene expression after 96-h Transwell co-culture with pooled donors of obASCs. PCR arrays were conducted with an *n* = 1 for each condition.

Gene Expression Changes after Transwell Co-Culture with obASCs
ER WT	ER MUT
Gene Name	Fold Change		Gene Name	Fold Change	
ABCB1	6.48	↑	ABCB1	4.99	↑
ADAM23	3.93	↑	CTNNB1	4.29	↑
ATM	0.3	↓	CTSD	0.41	↓
CCNA1	0.2	↓	MKI67	0.35	↓
CCND2	12.51	↑	MUC1	0.45	↓
CDH13	5.41	↑	NME1	0.34	↓
CDKN1C	0.41	↓	PTEN	2.36	↑
CDKN2A	12.51	↑	SERPINE1	4.47	↑
CSF1	3.72	↑	VEGFA	7.41	↑
CST6	5.6	↑			
ESR2	4.98	↑			
GLI1	2.47	↑			
GSTP1	2.51	↑			
HIC1	3.31	↑			
IGF1	5.22	↑			
IGFBP3	4.18	↑			
IL6	4.42	↑			
KRT5	12.51	↑			
MMP2	12.51	↑			
PGR	0.34	↓			
PLAU	3.82	↑			
PTGS2	12.51	↑			
PYCARD	0.42	↓			
RARB	0.38	↓			
SERPINE1	2.51	↑			
SFRP1	12.51	↑			
SCL39A6	0.42	↓			
SNAI2	0.41	↓			
TFF3	0.29	↓			
TGFB1	2.01	↑			
TWIST1	12.51	↑

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
