# Peer review of "Obesity-Altered Adipose Stem Cells Promote ER+ Breast Cancer Metastasis through Estrogen Independent Pathways"

_ijms, 2019, doi:10.3390/ijms20061419_

Round 1
Reviewer 1 Report
This study was aimed to evaluate the role of obASCs cells in the tumor growth and promotion of metastasis in different types of breast carcinoma experimental models. Authors have shown that obASCs promote tumor growth in ER wild type xenografts and promote metastasis in ER WT and MUT xenografts and ER MUT PDX models demonstrating that obASCs promotion of metastasis seems to be estrogen independent. Finally, the authors have found that obASCs significantly upregulate genes SERPINE1 and ABCB1 in ER WT and MUT breast cancer. All these findings are very important in view of the role of obesity and ER status in breast cancer. Methods are described in detail. A subscribed manuscript is capable of being published depending on minor revision process.
Minor points
1. Page 2, Introduction section: Authors should include the reason why exactly D451G mutation was selected for the study and describe its clinical significance similarly to Y537S mutation.
2. Page 2, line 66: Please specify the reference of CRISPR technology used for Y537S insertion.
3. Page 9, line 259-273:Please describe control group of PDX models used in this study.
4. Authors should describe all used statistical tests in the legends of Figures and significance after the post-hoc tests.
Author Response
1. Thank you for your comment and for catching this error. This was a typo WHIM43 has the D538G mutation and language has been added (lines 80-82) reflecting the constitutive ERa activity that results from both the Y537S and D538G mutations.
2. Language has been clarified and reference with CRISPR methodology has been inserted.
3. Language has been added to describe the control groups:” Control groups had PDX tumor coated in Matrigel and implanted bilaterally in the mammary fat pads under isoflurane and oxygen.”
4. Description of statistics has been added to figure legends
Reviewer 2 Report
It is well-designed study using established methods. It is useful article with original results.
I recommend the acceptance, however, several points need to be addressed.
1. Authors must implement several papers in the first paragraph of Introduction, which provide general information that metabolic reprogramming was proved to be effective against carcinogenesis in vivo. I strongly suggest to cite these papers:
1. Eur J Cancer Prev. 2010 Sep;19(5):379-84.
2. Int J Mol Sci. 2018 Oct 26;19(11). pii: E3335.
3. Int J Mol Sci. 2014 Jun 26;15(7):11435-45.
2. Fig. 1A, authors used symbol „#“ in the graph, but it is not explained in Notes. On the other hand, they mentioned symbol „***“ in Note of Fig.1, but it is not shown in graphs (similarly symbol „*“ in Fig.2).
3. The title of subchapter 2.3 is too long, I suggest to short it: „Regulation of breast cancer related genes in ER WT and ER MUT cells by obASCs“
4. Line171, please correct: „vinca alkaloids“
5. Authors used too many references older than 2010, it consists of more than 50 % of all citations, there are several suggestions to improve this ratio:
1. Oncogene. 2019 Feb;38(6):767-779.
2. Stem Cell Res Ther. 2017 May 25;8(1):121.
3. Oncol Lett. 2018 Feb;15(2):1403-1410.
4. Stem Cells. 2017 Sep;35(9):2060-2070.
5. Philos Trans R Soc Lond B Biol Sci. 2018 Jan 5;373(1737).
6. Eur J Cancer Prev. 2013 Jul;22(4):352-7.
I congratulate the authors to valuable article!
Author Response
1. Thank you for your suggestion. Discussion of metabolic reprogramming has been added.2. These changes have been made in the figure legends.
3. Thank you for the suggestions. This title has been changed to the suggested shortened version
4. This spelling error has been corrected
5. The above references were added to the manuscript with the exception of number 5 which evaluates the cisplatin-induced release of extracellular vesicles in ovarian cancer and was not found to be pertinent to the discussion or introduction of this manuscript.
Round 2
Reviewer 2 Report
Authors implemented all my suggestions. I recommend to publish this manuscript.